# The Role of Antibiotic Use in Third Molar Tooth Extractions: A Systematic Review and Meta-Analysis

**DOI:** 10.3390/medicina59030422

**Published:** 2023-02-21

**Authors:** Elham Torof, Hana Morrissey, Patrick A. Ball

**Affiliations:** School of Pharmacy, University of Wolverhampton, Wolverhampton WV1 1LY, UK

**Keywords:** antibiotics, adverse effects, dental infections, prophylaxis, treatment

## Abstract

*Background and Objectives*: Anecdotal evidence suggested variation in practices for antibiotic prescribing around dental procedures including route of administration of antibiotics, timing of the course prescribed (before, after or both), length of course prescribed, narrow vs. broad spectrum agents prescribed, use of single or combination of antibiotics, and the use of loading doses. This review aims to investigate this disparity of practices and the absence of global and local recent consensus on the most appropriate antibiotic interventions around invasive dental procedures. *Material and methods*: Following PRISMA-P^©^ methodology, a systematic review of randomised controlled clinical trials was designed, reviewed, and entered on the PROSPERO^©^ website prior to commencement. Ethics approval was gained from the University of Wolverhampton Committee. Searches were performed using PubMed^©^, Science Direct™, and the Cochrane Database, plus the bibliographies of studies identified. They investigated studies examining the efficacy and safety of any antibiotic regimen tested, independent of regimen used, versus a placebo, control, or no therapy, on outcomes in post third molar extraction. *Results*: The primary outcome of interest was postoperative infection and secondary outcomes were other post-surgical related complications of infectious nature and antibiotic adverse events. Sixteen RCTs were identified that met the selection criteria. Antibiotic use was reported to be safe, causing few adverse events. Meta-analysis of infection events showed antibiotics reduced the risk of an infection by 69%, but routine use for prophylaxis in uncomplicated procedures was not supported, and their role in patients with comorbidities or impaired immunity remains controversial. The effect on the incidence of dry socket showed no difference based upon regimen used. No significant benefit was found with respect to reduction of intraoral inflammation, wound dehiscence, haematoma, and lymphadenopathy. *Conclusion*: The effect on postoperative pain reduction was inconclusive. Routine use of antibiotics around M3 extraction procedures is not supported, but their use in the presence of co-morbidities and or immunosuppression remains controversial to be confirmed by future studies.

## 1. Introduction

Third molar teeth (M3s) are commonly known as ‘wisdom teeth’ and are the last teeth to erupt in the upper (maxillary) and lower (mandibular) jaws. They generally erupt between the age 18 and 24, which has been referred to as ‘the age of wisdom’ [1]. There are four ‘wisdom teeth’: upper left, upper right, lower left, and lower right. The tooth most commonly impacted is the lower M3 and extraction of one that is impacted is more likely to be associated with surgical complications due to their abnormal position and blocked eruption, leading to recurrent third tooth pericoronitis [2].

The overall incidence of impaction of M3s ranges from 36% to 59% [3], with approximately 60,000 extracted per annum in secondary care in the UK [4]. The incidence varies across different populations and ethnic groups [5].

While not every asymptomatic or pathology-free impacted M3 causes a clinical problem, the National Institute for Health and Care Excellence (NICE) recommends that pathological M3s should be removed [6]. Removal of asymptomatic and/or pathology-free M3s is termed prophylactic.

Early diagnosis and treatment reduce postoperative complications that typically include pain and swelling resolving within a few days. Other potential complications include trismus, haemorrhage, localised alveolar osteitis, and/or dry socket, damage to the inferior alveolar or lingual nerves, and periodontal damage [7]. Variables contributing to the incidence and severity of postoperative complications include the duration of surgery and surgical techniques implemented, including triangular flap design and irrigation method [8].

Several classification systems estimate the surgical difficulty of M3s removal and help to determine the best methodology for extraction, based on preoperative assessment of panoramic radiographs.

Pathology from M3 impaction is multifactorial, including spatial relationship, type, level and depth of impaction, angulation of the M3s, number and shape of the roots, ramus relationship, and space available. The difficulty assessment uses the Pederson Difficulty Index (PDI) [9]. Winter’s classification is commonly used for spatial assessment of impacted teeth [10].

Apart from radiographic assessment, previous studies have suggested that surgical technique and its difficulty, together with the experience of the surgeon, are further predicators of difficulty [11,12].

The human oral cavity contains multiple microorganisms, microbial habitats such as teeth, gingival sulcus, attached gingiva, tongue, cheek, lip, hard palate, and soft palate. Microorganisms from the oral cavity can cause a range of oral infections, and the tissue trauma associated with tooth extraction causes bacteraemia [13].

An antibiotic used for prophylaxis should be bactericidal against the most common microorganisms that cause infection. In oral surgical procedures these are *Staphylococcus* spp., *Streptococcus* spp., and anaerobic Gram-positive and Gram-negative rods [14]. The prescription is usually given for a short period of time, typically no more than 7 days. Antibiotics (ABs) should be used to reduce the infection rate, but their unnecessary use may have serious adverse drug events, including allergic reactions, bacterial resistance, and risk of *Clostridium difficile* infection (CDI) [15].

The main purpose of pre-operative use of AB is to provide an adequate drug level in the exposed and damaged tissues before, during, and for the shortest possible time after the operative procedure.

The review question was formulated as: What is the scientific evidence available to support the use of AB to prevent infection and its complications after M3 tooth extraction procedures?

### Review Rationale

This project was developed based upon anecdotal evidence of a wide variety of practices for antibiotic prescribing around dental procedures including route of administration of antibiotics, timing of the course prescribed when invasive procedures are planned (before, after, or both), length of course prescribed, narrow vs. broad spectrum agents prescribed, use of single or combination of antibiotics, and the use of loading doses. Additionally, there is a disparity of which (if any) antibiotic intervention is more effective than no intervention at all, or for which patients they should be prescribed to. This project attempted to investigate this disparity of practices and the absence of global and local recent consensus on the most appropriate antibiotic interventions around invasive dental procedures.

## 2. Materials and Methods

Ethical review was undertaken, and approval granted by the University of Wolverhampton Life Sciences Ethics Committee LSEC/202021/HM/9. The aim of the study is to ascertain if there is scientific evidence to support whether AB can effectively reduce the postoperative infections after M3 tooth extraction. To achieve the aim, the following aspects were investigated: The regimen of ABs commonly used in M3 extraction procedures, the efficacy of AB in preventing infection and its complications after M3 extraction, and the safety of using AB used in M3 extraction procedures.

The study was designed in accordance with the ‘Preferred Reporting Items for Systematic Reviews and Meta-Analyses’ (PRISMA-P^©^, Ottawa Hospital Research Institute and University of Ottawa, Ottawa, Canada) guidelines and adhered to the Cochrane Handbook for Systematic Reviewers’ methodological guidelines [16]. This systematic review was designed prior to commencement and registered into the PROSPERO^©^ (National institute of health, York, UK) website platform; reg. no CRD42021269522. Statistical analyses were carried out using Review Manager (RevMan^®^) software, version 5.4.1 (available from The Cochrane Collaboration, Oxford, UK). The Cochran–Mantel–Haenszel (M-H2) was used for combining dichotomous data, and since all variables were dichotomous, the effect measures estimated were relative risk (RR), which was reported with 95% confidence intervals (CIs).

Each study included in the metanalysis was estimating a different effect size with respect to the patient population, control interventions used, and varying dosages of interventions. The statistical unit was the participant. The measure of effect size was based on ratios found in the study. For outcomes that were statistically significant, the number needed to treat (NNT) was calculated to estimate the overall clinical impact of the intervention. All decimals in the NNT were rounded to the nearest whole number. The statistical significance of the overall result is also expressed with the probability value (*p* value) with significance regarded as *p* < 0.05.

Heterogeneity of included data was measured by Chi^2^ test by dividing the result of Cochran’s Q test and its degrees of freedom by the Q-value itself. Significant heterogeneity was defined as chi-square test with *p* < 0.1 or *I*^2^ statistics >50%. *I*^2^ described the percentage of variability in results across studied in the meta-analysis that is due to real differences rather than chance [17].

Sensitivity analysis was used when *I*^2^ was more than 50%, removing each trial one by one to evaluate the stability of the results.

### 2.1. Search Strategy

Medical Subject Headings (MeSH) were applied and the following selected key words were used in various combinations in PubMed database: (“dental extraction” OR “dental extractions” OR “extractions” OR “extraction” OR “third molar” OR “third molar surgery” OR “dental procedure” OR “third molar removal”) AND (“Antimicrobial” or “antibiotics”, “prophylactic” OR “prophylaxis” OR “pre-operative” OR “preoperative” OR “peri operative” OR “postoperative” OR “preventative”) AND (“efficacy” OR “effectiveness” OR “effect” OR “effective”. The following filters, as key words for titles and abstracts, were also applied: randomised controlled trial OR randomised trial OR clinical trials. The date of last search was 30 November 2021.

Studies were assessed for eligibility against the following criteria developed using the PICOS tool (population, intervention/investigation, comparators, outcomes, and setting):(P):Types of participants: Adult healthy patients of any age and gender undergoing uncomplicated single or multiple tooth extraction of any tooth.(I):Types of interventions: RCTs where oral ABs, independent of the type of ABs used, dose administered, timing of course administration (preoperatively, postoperatively, or both), number of doses administered, and length of the regimen.(C):Comparing all possible monotherapy AB treatment with either placebo or control or no treatment or no AB.(O):Type of outcome measures: Primary outcome of interest was postoperative infection. Secondary outcomes: other post-surgical related complications of infectious nature (e.g., alveolar or alveolitis osteitis or dry socket, pain, wound dehiscence, swelling, temperature), AB adverse events (e.g., gastric complications, stomach pain, diarrhoea, headache).(S):Primary care, community, or hospital setting.

### 2.2. Selection Criteria

Studies were included in the review when they were:
RCTs, double-blind and placebo-controlled clinical trial.Patients undergoing dental extraction.Only oral or systemic route of AB administration.Studies comparing the use of any AB (independent of the type of ABs used), dose administered, timing of course administration (preoperatively, postoperatively, or both), as part of treatment versus a placebo alone or control or no AB following M3 tooth extraction procedures.Adult over 17 years of age.Studies published from January 2000 to November 2021.

Studies were excluded from the review when they were:Systematic and meta-analysis review or literature review study.Non-human or animal study.Published in language other than English, where full translated version could not be supplied by the publisher or the author/s.Abstracts and conference proceedings.Trials which are comparing AB versus non-AB agent/s.Unhealthy patients or patients with comorbidities which put them at high risk of infection.Trials comparing AB versus another AB/s without a placebo arm.Wrong formulation (injections/gels).

### 2.3. Selected Studies Summary

Following the Cochrane handbook [18], the quality of each RCT was determined using The Effective Public Health Practice Project (EPHPP, EPHPP McMaster University 1280 Main Street West, Hamilton, Ontario, Canada, L8S 4K1) quality assessment tool rating for individual studies [19]. A flow diagram of the included study population (patients) was reported according to Consolidated Standards of Reporting Trials (CONSORT, The EQUATOR Network and UK EQUATOR Centre, Centre for Statistics in Medicine, Nuffield Department of Orthopaedics, Rheumatology and Musculoskeletal Sciences, University of Oxford, Botnar Research Centre, Windmill Road, Oxford, OX3 7LD, UK), and a PRISMA chart was used to present the final included studies. Publication bias was also calculated using RevMan^©^ software version 5.4.1.

Based on the study selection criteria, a total of 494 articles were identified in the literature search. The title and abstract screening was conducted independently by the three authors, a final list of 64 articles was selected, and 48 were excluded for various reasons. Following full-text review by the three authors, 16 RCTs met the criteria for inclusion in this systematic review.

Two RCTs were included as narrative in the systematic review [20,21]. Fourteen RCTs were included in the meta-analysis [22,23,24,25,26,27,28,29,30,31,32,33,34,35].

Figure 1 is the PRISMA flowchart of the study selection process. The flow diagram of the included study population according to Consolidated Standards of Reporting Trials (CONSORT) shown in Figure 2.

Further to that, the researcher sought advice from three dentist academics with systematic review experience regarding the review direction, and later compared the findings with previous published systematic reviews.

## 3. Results

RCTs fulfilling the inclusion criteria were assessed for quality by two reviewers [18]. Table 1 and Table 2 describe the characteristics of the included studies and the reviewer comments. The EPHPP tool [19] was used to examine each study against six components. It was judged to be suitable to use to evaluate the effectiveness of interventions in systematic literature reviews [18]. Table 3 shows the global quality rating of the selected studies. All studies were classed as moderate–strong global rating on the EPHPP.

### 3.1. Statistical Analysis

#### 3.1.1. Analysis of Clinical Trials with Infection as the Outcome Measure

Twelve studies investigated the international guideline on the efficacy of AB on reduction of infection following M3 dental extraction [22,23,24,25,27,28,29,30,31,32,34,35]. A total of 2069 patients were randomised to AB or placebo group in 12 RCTs. Infection occurred in 34 of 1186 patients receiving AB and 78 of 883 patients on placebo or no treatment.

The overall combined effect shows that AB, independent of the type, dose, timing of administration (preoperatively, postoperatively, or both), were beneficial (*p* < 0.001).

Of the RCTs, 12 [22,23,24,25,27,28,29,30,31,32,34,35] identified the number of patients with infection, 7 [24,25,26,27,28,32,33] investigated the number of patients with dry socket, and 7 [26,28,29,30,31,34,35] reported total adverse events that occurred between groups during the study period. A funnel plot generated to test for publication bias showed no asymmetry (Figure 3 and Figure 4).

#### 3.1.2. Analysis of Clinical Trials with Dry Socket or Alveolar Osteitis or Alveolitis as the Outcome Measure

In 7 trials [24,25,26,27,28,32,33], 22 patients developed dry socket (Figure 5). The 95% confidence intervals of the overall effect estimate the effect line, indicating no significant difference in dry socket events. Due to the small number of studies included, a funnel plot was not assessed [35].

#### 3.1.3. Analysis of Clinical Trials with Medication Related Adverse Events as the Outcome Measure

Eight studies [26,27,28,29,30,31,34,35] analysed the influence of AB on the incidence of adverse events. Of 148 adverse events reported, 88 occurred in AB groups and 60 in placebo/no AB groups. The random effects model summary result of RR yielded 0.96, confidence interval (0.58, 1.59), showing no significant difference (Figure 6).

### 3.2. Narrative Analysis of other Outcomes

#### 3.2.1. Increased the Risk of Infections

Milani et al. [22] evaluated local infection dichotomously (presence = 1 and absence = 0) by clinical examination. It was found in four patients (one in group-1, three in group-2, and zero in group-3), which was not significant. Mariscal-Cazalla et al. [31] reported that no significant differences (*p* = 0.064) in infection rate were found between pre- and postoperative AB and postoperative AB. Rescue AB use after surgery was assessed based on need. More controls required rescue AB therapy (*p* = 0.013) than in AB-treated groups. Lacasa et al. [35] found a higher rate of infection (*p* = 0.006) in patients receiving placebo (16%) than those receiving single-dose prophylaxis or 5-day pre-emptive therapy (5.3% and 2.7% respectively). Sekhar, Narayanan, and Baig [23] reported only one purulent wound in their preoperative metronidazole 1 g groups versus none in the placebo group. No significant difference was reported in wound infections (*p* = 0.40).

Ataoğlu et al. [32] reported that six preoperative AB patients had early wound infection and there were seven in the placebo group. Late wound infection was reported in one patient with pre-operative AB and one patient in postoperative AB, with two late wound infections observed in their controls, which was not significant.

Siddiqi, Morkel, and Zafar [24] reported that 6 sockets from 380 patients became infected (4 in the placebo group 2 in the AB group), also not significant. Arteagoitia et al. [29] reported infections in five patients in the placebo group, all in the first postoperative week, and in two in the AB group, both > 1 week post-op (*p* = 0.278). Bezerra et al. [25] observed surgical wound infection in three control patients (4.14%), with no significant difference compared with the experimental group.

Xue et al. [27] had two wound infections (1%) each among their AB and placebo group. Arteagoitia et al. [34] reported inflammatory complications (IC) in 35, 30 in placebo (12.9%), 5 in AB group (1.9%). The incidence of ICs was between 2.9 and 19.9 times more frequent if postoperative AB were not taken (*p* = 0.001).

Luaces-Rey et al. [28] found post operative infection two patients in group-1 (2.9%) and three patients in group-2 (4%). One patient needed hospital admission for intravenous AB treatment. López-Cedrún et al. [30] reported five placebo group patients with infected socket after surgery (*p* = 0.001). In the study by Lacasa et al. [35], the incidence of infection was higher in the placebo group than the five-day pre-emptive therapy and those receiving single-dose prophylaxis: 16% (12/75), 2.8% (2/72), and 5.3% (4/75, which was significant, *p* = 0.014).

Lacasa et al. [35] found the risk of infection in cases of difficult surgery (ostectomy) was reported as 24%, 9%, and 4% with placebo, prophylaxis, and pre-emptive treatment, respectively, and the risk of infection in cases of ostectomy surgery was observed in two patients in prophylactic and two patients in pre-emptive therapy and ten patients in placebo group with significant differences found between groups (*p* = 0.011); whereas the risk of infection in cases when ostectomy was not performed was 7%, 2%, and 1% with placebo, prophylaxis, and pre-emptive treatment, respectively, which was not significant (*p* = 0.339). Lacasa et al. [35] affirmed that difficulty of surgery increases the incidence of infections (*p* = 0.027).

The logistic regression was carried out by Lacasa et al. [35] using duration and difficulty of the surgical procedures as potential dependent variables on risk of infection and they concluded that while the pre-emptive therapy group had a slightly greater length of procedure, it did not achieve significance. Moreover, Lacasa et al. [35] found that the reported interventions lasting less than 5-min had an incidence of infection of 1.6% (1/64), whereas the duration of surgery lasting 5–10 min, the incidence of infection of increased to 7.4% (6/81) and with interventions lasting more than 10 min, the incidence of infection increased to 13.8% (11/80). Similarly, the study by Luaces-Rey et al. [28] found correlation between timing of surgery and risk of infection. Two patients with surgical infection were recorded in the less than 5 min procedure group, two patients in the group between 5 and 10 min, and one patient in a surgical extraction longer than 10 min. Based on the results of the study, there was no statistically significant correlation observed between the treatment group and timing of the surgery. Arteagoitia et al. [34] observed ages as a confounding variable (*p* = 0.029) and reported that as the patient’s age increases, the possibility of risk of inflammatory complication increases by 1.08 per year of age, whereas the possibility of postoperative complications is 10% without AB at 20 years and exceeds by 30% at 40 years. Similarly, López-Cedrún et al. [30] found that age, gender, and operative time have a significant effect (*p* = 0.047) on incidence of infection and concluded that infection rate was higher in the women participants and older participants than younger patients (25.2 vs. 22 years). Moreover, the risk of infection was influenced by operative time in which mean operative time was longer in the infected patients (277 s) than patients without infection (239 s).

Age, length of procedures, and placebo use were the three factors associated with most infections (Table 4). Four studies (19%) did not report on increased infection [20,21,26,33].

#### 3.2.2. Antibiotic Related Adverse Events

In the study by Mariscal-Cazalla et al. [31], no significant differences in adverse effects of medications were observed between groups. Adverse effects were reported in four participants (one event in pre- and post-operative group, one event in postoperative group, and two events in placebo group); however, the types of events were not described. Lacasa et al. [35] reported that adverse events occurred during therapy and for 30 days afterwards. Diarrhoea was reported by three patients in placebo group and nine patients in the prophylaxis group. Only two cases of severe diarrhoea (one in the placebo group and one in the prophylaxis group) led to participant withdrawal and discontinuation of the study medication. The number of patients with headache was 14 cases in the placebo and 17 cases in the prophylaxis groups. The study by Arteagoitia et al. [29] reported that adverse events were more frequent in the AB group, where four mild events were reported (*p* = 0.009). Cases of nausea and vomiting (*n* = 1), diarrhoea (*n* = 8, 1 case in placebo group), abdominal pain (*n* = 1), and vaginal candidiasis (*n* = 1) were reported. Kaczmarzyk et al. [26] ceased clindamycin in three patients in the AB group from day 5, due to gastric complications (stomach pain and diarrhoea) and replaced it with metronidazole orally at a dose of 500 mg three times a day.

The study by Xue et al. [27] reported 14 adverse events in the AB groups: gastrointestinal (*n* = 4), bleeding (*n* = 2), ulcer (*n* = 2), and fever (*n* = 6). In the placebo group, there were 22 adverse reactions, reported as: bleeding (*n* = 6), ulcer (*n* = 2), and fever (*n* = 14). There was no significant difference between the two groups. Arteagoitia et al. [34] reported 16 mild cases,14 in AB and 2 in placebo (2 vomiting events, 2 gastric pain events, 1 mycosis event, and 11 diarrhoea events) were reported and with some participants admitted to hospital. Adverse events, expectedly, were more frequent in the AB group (*p* = 0.009).

Luaces-Rey et al. [28] reported diarrhoea in group-1 (*n* = 1) and group-2 (*n* = 1), nausea or vomiting in group-2 (*n* = 2), and epigastralgia in group-1 (*n* = 1). The study by López-Cedrún et al. [30] reported 13 adverse events in their preoperative group, 14 cases in placebo group and 10 cases in postoperative group. Adverse events included vomiting (*n* = 1), nausea (*n* = 2 in AB and 2 in placebo groups), diarrhoea (*n* = 3 in AB and *n* = 1 in placebo groups), gastric pain (*n* = 5 in AB and *n* = 1 in placebo groups), rash (*n* = 2 in AB group), headache (*n* = 2 in AB and *n* = 3 in placebo groups), and other (*n* = 8 in AB and 7 in placebo groups). The author concluded that the reported adverse events might be related to the administration of ibuprofen rather than the ABs. No significant differences in adverse events were found between groups. Comparing the number of events between AB arms and placebo was not significant with *p* = 0.725 (Pearson test). Eight studies did not report adverse events [20,21,22,23,24,25,32,33].

The study by Milani et al. [22] found no differences among the three groups at day 4 and 7 after surgery (*p* = 0.060, *p* = 0.330 respectively). Mariscal-Cazalla et al. [31] used patients’ self-evaluated postoperative swelling of the treated area as endpoints 7 days after extraction. The placebo group reported significantly greater swelling values than AB prophylaxis at 48 h (*p* = 0.012); 72 h (*p* = 0.001); and 1 week (*p* = 0.02) following M3s removal. However, Lacasa et al. [35] reported no significant difference between the AB prophylaxis group (mean 1.24) and AB pre-emptive course group (mean 2.68) groups.

In the study by Sekhar [23], swelling was measured using four grades (none, mild, moderate, severe); moderate swelling was reported in two patients in the placebo group, one patient in preoperative metronidazole 1 g, and no cases reported in 5-days postoperative metronidazole 400 mg. The degree of swelling was significantly less in the 5-day group (*p* = 0.03). Ataoğlu [32] reported no significant differences between groups. Additionally, Siddiqi [24] measured facial swelling using clinical observation after 3 days, 7 days, and 2 weeks postoperatively. They reported no significant differences (*p* > 0.050). The study by Kaczmarzyk et al. [26] found no significant difference between groups on days 1, 2, and 7 (*p* = 0.5, 0.61, and 0.4, respectively). Xue et al. [27] also reported no significant difference between groups (*p* = 1.000). The study by López-Cedrún et al. [30] reported 35 patients had intraoral swelling at day 7, with 10 in group A, 12 in group B, and 13 in group C, differences were not statistically significant (intraoral *p* = 0.871, extraoral *p* = 0.588) between the groups after surgery.

Examining the combined 16 studies, only two studies [23,31] found the use of ABs favourable (Table 5). Seven (44%) studies did not report on postoperative facial oedema or swelling [20,21,25,28,29,33,34].

### 3.3. Risk of Bias

RCTs’ RoB was assessed using the Cochrane risk-of-bias version 2 (RoB-2) tool [36,37]. Seven domains were analysed including selection bias, performance bias (including blinding personnel and participants), detection bias (blinding outcome assessment), attrition bias, reporting bias, and method of randomisation. RoB was categorized as low, high, or unclear. Among the sixteen included RCTs assessed, one trial [32] was rated high RoB in one or more key domains, mostly due to unclear random sequence generation, allocation concealment, incomplete outcome reporting, and the lack of blinding of assessors, while the remaining fifteen had a low RoB (Figure 7 and Figure 8). Risk of bias assessment was conducted independently by the three authors and the final verdict was made in a face-to-face meeting.

Selection bias: Fifteen RCTs adequately described the generation of random sequence, but one did not provide a description [32].

Allocation concealment: Similarly, only one trial did not clearly describe allocation concealment [32].

Performance bias: Concerning performance bias, four studies were considered unclear due to insufficient details of the assessors blinding [20,21,32,35]. One trial provided no information on whether patients were blinded [32].

Detection bias: Eleven studies had no description of blinding of the outcome assessment resulting in unclear detection bias. However, five studies [21,22,23,29,31] were judged to have low RoB for detection bias domain.

Attrition bias: Most studies provided sufficient information concerning the reasons for exclusion, apart from two studies [27,32] judged to have an unclear RoB.

Reporting bias: Two trials appeared to have a high RoB due to missing outcome data for the selective reporting domain. Lacasa [35] was judged to have a high RoB due to microbiological isolation, and identification was not reported. For Bezerra [25], body temperature was not reported.

Other bias: No other important sources of bias were identified. Therefore, all trials were judged as low risk.

### 3.4. Number Needed to Treat 

1.To prevent infection events

Control/placebo group event rate (CER): 78/883 = 0.0883352

AB group event rate (EER): 34/1186: 0.0286677

Absolute risk reduction (ARR) = 0.0596675

Number needed to treat (NNT) = 1/ARR = 1/0.0596675 = 16.759542

Approximately 17 patients undergoing dental extraction need to receive AB to prevent one infection from occurring. This was smaller than that reported by Lodi et al. [38] and Ren and Malmstrom [39] as 19 to 25. This review included more recent studies which were not included in Lodi et al. [38] and Ren and Malmstrom [39], which might have impact on the lower NNT in this review.

NNT is more than 5 which indicates that the intervention (use of prophylactic Abs) is not effective in preventing infection [40,41].

2.To prevent dry socket events

Control/placebo group event rate (CER): 24/485: 0.0494845

AB group event rate (EER): 22/562: 0.0391459

Absolute risk reduction (ARR) = 0.0103386

Number needed to treat (NNT) = 1/ARR = 96.72

Approximately 97 patients undergoing dental extraction need to receive Abs to prevent one dry socket from occurring which is higher than that reported by Ren and Malmstrom (2007) and Lodi et al. (2021) as estimated to be between 13 and 46. NNT is more than 5 which indicates that the intervention (use of prophylactic Abs) is not effective in preventing infection [40,41].

3.To cause adverse events

Control/placebo group event rate (CER): 95/728 = 0.1304945

AB group event rate (EER): 139/935 = 0.1486631

Absolute risk reduction (ARR) = 0.0181686

Number needed to harm (NNH) = 1/ARR = 1/0.0181686 = 55.56

Approximately 55 patients undergoing dental extraction need to received AB to cause one adverse event such as diarrhoea, hypersensitivity, and thrombocytopenia. NNH is not a negative figure but is more than 5 which indicates that the possible harm caused by the use of ABs exists, but it is very small and does not negate the AB use when required [40,41].

### 3.5. Limitations

The research was low-cost; as such, only databases accessible through the researcher’s institution were possible to use. Authors and publishers were approached to provide English versions, complete text, or additional data; however, in the cases where response was not received or received informing of unavailability, the study could not be included. It was not possible to pay for translating text. The major factors that may have had an influence on the infection outcomes were not reported in all studies, and when they were reported, there were some variations on how they were measured. Thus, it was not possible to determine to what extent ABs may have contributed to the reduction of infection rate and its related complications.

The AB most used was amoxicillin 1 g; however, there was insufficient evidence to confidently conclude that it is the most effective AB or which dose should be recommended for prophylaxis or pre- and postoperative preventative therapy. However, all used AB protocols had the tendency to reduce infection rate. In addition, RCTs in this metanalysis were conducted in various countries within different healthcare systems and governed by variable guidelines. Therefore, drawing definitive conclusions about the effects of AB is problematic. Moreover, there was great variability in how the variables in 6.2.1 and 6.2.2 were defined and measured, e.g., teeth in different positions, depth and angulation of the molar to be extracted. This systematic review looked at variation in outcomes in healthy patients and not patients with physical long term and chronic diseases (infection high-risk patients).

## 4. Discussion

All studies used in this systematic review were randomised, double-blinded controlled trials, comparing different doses of AB with placebo or no AB use. All studies identified had M3s extraction performed by a single maxillofacial surgeon or specialist in oral and maxillofacial surgery with varying years of experience in dentoalveolar surgery under local anaesthesia. All postoperative evaluation was performed by a single examiner at scheduled times. However, some studies used patients’ self-evaluation to register follow-up examination parameters such as pain, trismus, body temperature, and number of analgesic pills taken daily. In all included studies except one, analgesics or anti-inflammatory drugs were used. Most studies included a variety of reasons for M3s extraction, but fully impacted teeth were presented in most studies and indication for extraction was homogenous among studies.

Three studies included a single ethnicity such as Caucasian, Chinese, or Latin American, which may make their results not globally applicable. Similar baseline characteristics of patients were reported between trials of adults >17 years of age with a slight female predominance in some trials. No RCTs included children, patients with infective endocarditis, or immune compromised patients.

The surgical technique was standardised in all studies; however, the surgical technique was not described in two trials. The design of the RCTs was described in all studies except one trial in which study design remained unknown.

There were differing practices with respect to time of ABs administration. Studies have investigated the use of ABs either several days preoperatively, as single dose, or postoperatively or in a combination with varying time windows of exposure to ABs, ranging from 2 days to 7 days. It has to be emphasised that all included trials received various combinations of different AB agents.

This review shows that the use of AB (in different regimens and types) can reduce infection and its complications when compared to individuals who did not use AB. In this analysis, factors such patients age, gender, extraction procedures length, and operator experience were considered, with the intention to explore evidence which may confirm, or deny, that the use of AB (in any dose or regimen) is an independent factor which can lead to reduction in infection rate after M3 extractions in healthy people. AB was found to be safe, causing few adverse events, which were deemed as insignificant compared to placebo. Meta-analysis of infection events showed that ABs reduced the risk of infection by 69% in the intervention groups. The efficacy of AB regimens on the reduction of dry socket suggests that there may not be a difference in dry socket outcomes between the regimens assessed; however, further research is required to confirm these results. Further analyses did not demonstrate statistically significant benefits to AB treatment in reduction of intraoral inflammation, wound dehiscence, lymphadenopathy, postoperative facial oedema, trismus, or postoperative pain. Similarly, there was no statistically significant difference in body temperature between groups when AB were administered. Two earlier systematic review/meta-analyses were found and compared to the results of this review to validate our findings (Table 6).

## 5. Conclusions

This review meets level A evidence criteria (multiple RCT double-blinded studies). In the M3 extraction meta-analysis, 16 RCTs were identified and analysed. Antibiotic use was reported to be statistically significant in preventing infection (*p* < 0.01). Prevention of complications such as dry sockets was not statistically significant (*p* = 0.34) and the NNT was larger than 5 (17 and 97, respectively), which indicates that the intervention not sufficiently effective to justify its use. The occurrence of side effects was not statistically significant (*p* = 0.88), NNH was 55 which indicates that the possible harm caused by the use of AB exists, but it is very small and does not negate the AB use when required. The results from this review were broadly consistent with previous systematic reviews. Standardised techniques and procedures, reasons for extractions, and adverse events reporting are required to enable further progress. Future reviews require sub-group analysis on specific populations including those with co-morbidities or existing immunosuppression to close a current gap in the available literature. It is worth noting that prevention of antimicrobial resistance may outweigh the finding of clinical significance, and future studies need to report data on mitigations made to prevent microbial resistance.

## Figures and Tables

**Figure 1 medicina-59-00422-f001:**
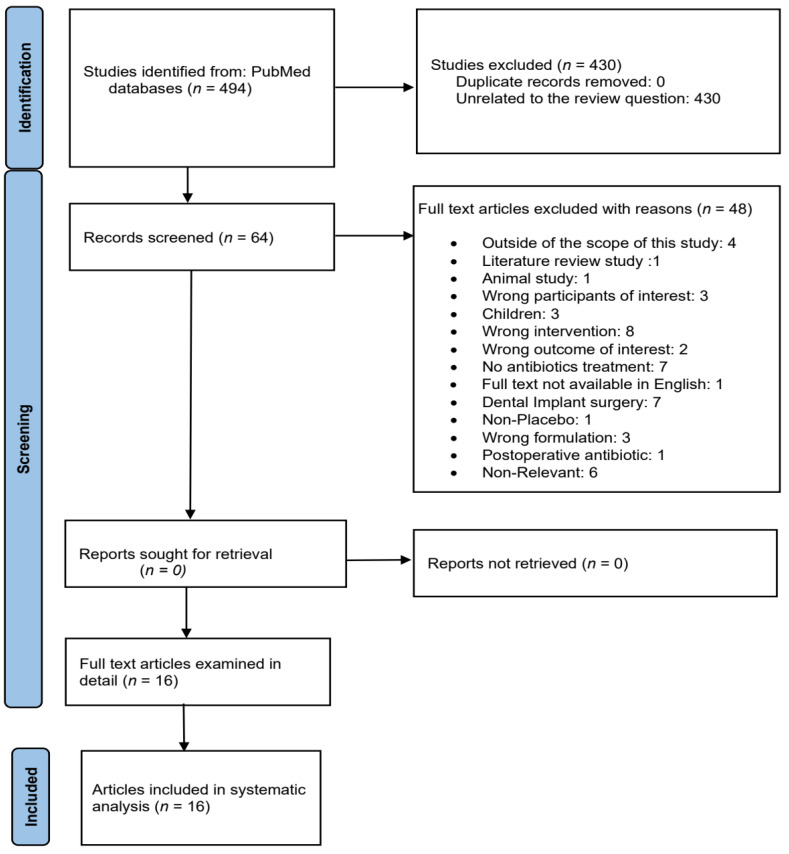
Flowchart showing the study selection process.

**Figure 2 medicina-59-00422-f002:**
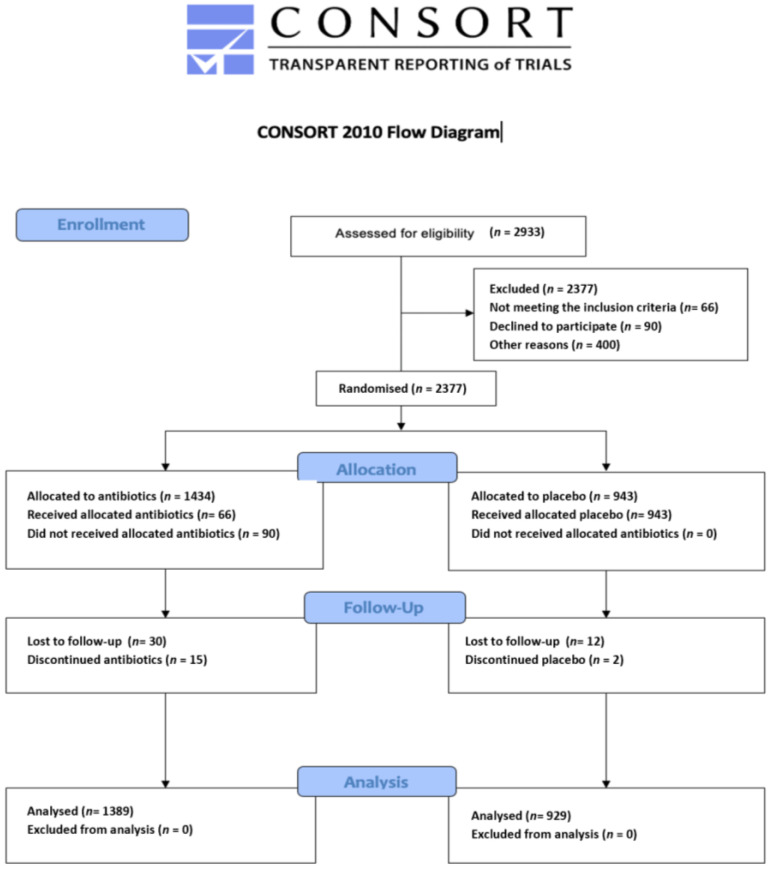
CONSORT flow diagram initial search.

**Figure 3 medicina-59-00422-f003:**
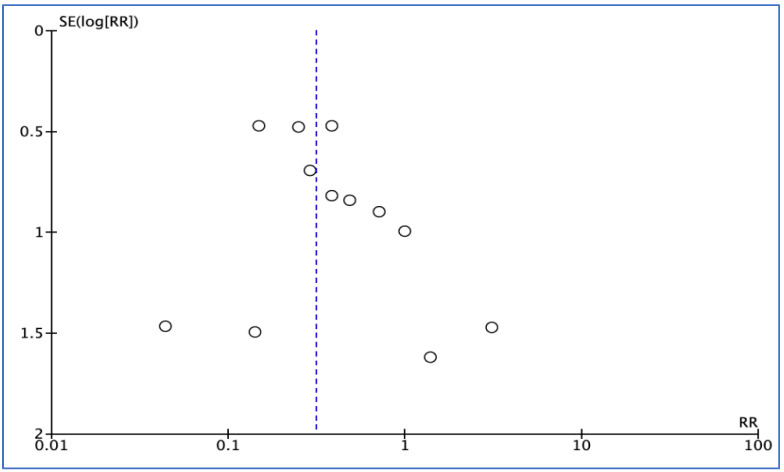
Funnel plot for infection outcome.

**Figure 4 medicina-59-00422-f004:**
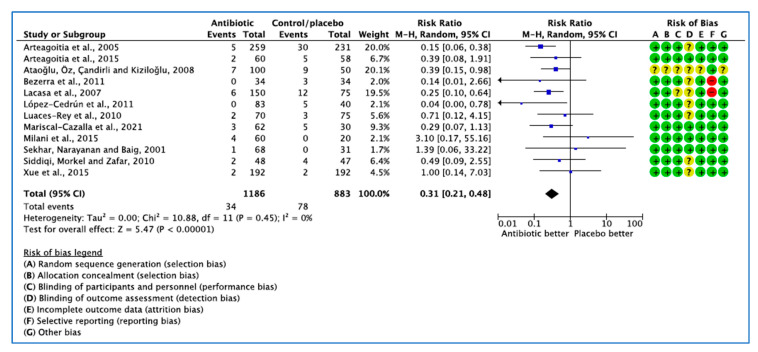
Forest plot of comparison between AB vs. placebo that gives summary estimate (centre of diamond) and its 95% confidence interval (width of diamond) for the infection outcome. Statistical method: Mantel–Haenszel with random-effects model. Events in column-1 are the number of patients with infection in AB and events in column-3 are the number of patients with infection in placebo group. The total column is the total number of patients allocated to the AB group and the total number of patients allocated to the placebo group.

**Figure 5 medicina-59-00422-f005:**
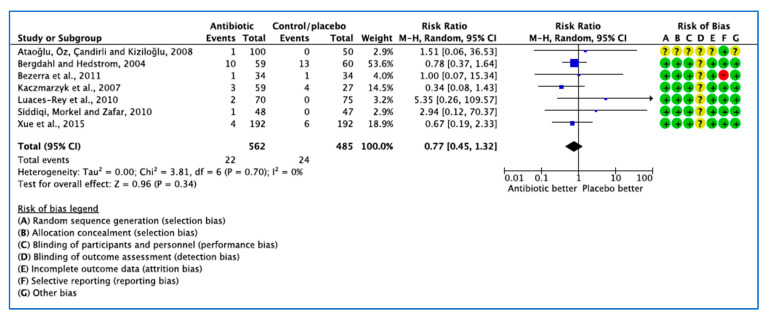
Forest plot of comparison between AB vs. placebo that gives summary estimate (centre of diamond) and its 95% confidence interval CI (width of diamond) for the dry socket outcome. Statistical method: Mantel–Haenszel with random-effects model. Events in column-1 are the number of patients with dry socket in AB and events in column-3 are the number of patients with dry socket in placebo group. The total column is the total number of patients allocated to the AB group and the total number of patients allocated to the placebo group.

**Figure 6 medicina-59-00422-f006:**
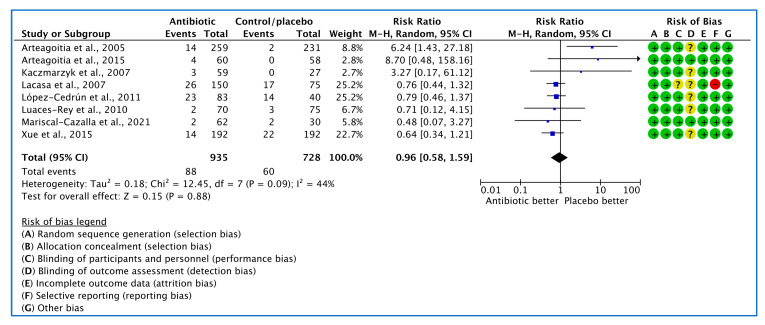
Forest plot of comparison between AB vs. placebo that gives summary estimate (centre of diamond) and its 95% confidence interval (CIs) (width of diamond) for the adverse events outcome. Statistical method: Mantel–Haenszel with random-effects model.

**Figure 7 medicina-59-00422-f007:**
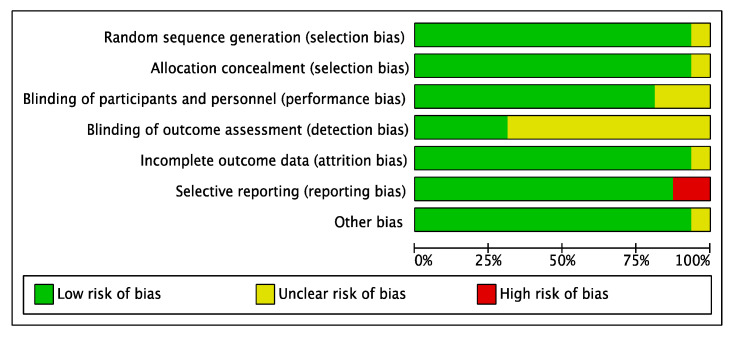
Risk of bias, review of authors’ judgments about risk of bias items for each included study.

**Figure 8 medicina-59-00422-f008:**
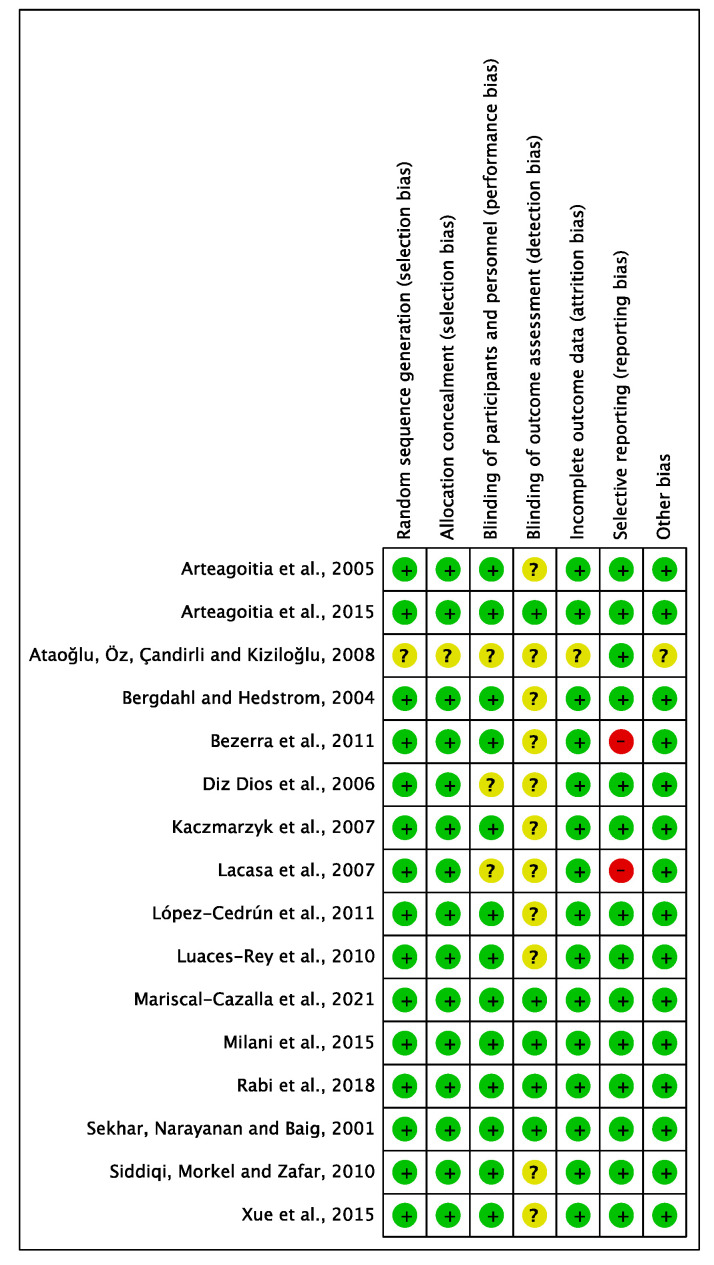
Risk of bias summary: each risk of bias item for each study included in the review (Arteagoitia [34], Arteagoitia [29], Ataoğlu [32], Bergdahl [33], Bezerra [25], Diz Dios [20], Kaczmarzyk [26], Lacasa [35], López-Cedrún [30], Luaces-Rey [28], Mariscal-Cazalla [31], Milani [22], Rabi 2018 [21], Sekhar [23], Siddiqi [24], Xue [27]).

**Table 1 medicina-59-00422-t001:** Main characteristics of the clinical trials included in the systematic review.

Citation	Design	Sample Age Gender	Study Objectives	Antibiotic	Control	Antibiotic Dosing	Mouthwash Use	Inflammation and Pain Relief	Extraction Reason	Follow Up	Study Period
Milani et al., 2015 [22] Europe	Double-blind RCT	*n* = 80 18–30 years. F: 54 M: 26	Evaluate the presence of mouth opening (trismus) in postoperative period	1 g amoxicillin + 500 mg 8 hourly for 7 days 1 g amoxicillin + placebo 8 hourly for 7 days	Placebo 1 h before surgery and 500 mg 8 hourly for 7 days.	1 h before surgery	N/R	Postoperative: For sever pain: 30 mg codeine + paracetamol 6 hourly as required/For less severe pain Metamizole + Adiphenine + Promethazine	Extraction of fully impacted lower 3Ms	7 days	Jan 2011–Jan 2012
Mariscal-Cazalla et al., 2021 [31] Europe	Double-blind RCT	*n* = 92 18 to 63 years. F: 55 M: 37	Evaluate the effectiveness of antibiotic therapy against possible complications (pain, inflammation, infection) of impacted mandibular third molar extraction	G1: (pre+ post antibiotic): 750 mg amoxicillin every 8 h for 2 days before surgery + for 5 days after surgery G2: (post antibiotic): 750 mg oral amoxicillin every 8 h for 5 days after surgery	Methylcellulose Tab for 2 days before surgery + for 5 days after surgery	2 days before surgery	Preoperative: Chlorhexidine mouthwash for 2 min Postoperative: Chlorhexidine mouthwash after brushing for 1 week after surgery for 7 days	Postoperative: Ibuprofen 600 mg 8 hourly for 2 days after surgery; If not adequate: 1 g paracetamol	Extraction of impacted lower 3Ms	7 days	Jan 2018–Dec 2018
Lacasa et al., 2007 [35] South Africa	Double-blind RCT, Parallel group phase III	*n* = 225 ≥18 years. M: 96 F: 129	Evaluate the efficacy of two SR antibiotic regimens in the reduction of infection after third molar extractive surgery	Prophylaxis G: 2 single dose Amoxicillin/clavulanate 1000/62.5 mg tab before surgery+ two placebo 2000/125 mg, twice daily for 5 days Pre-emptive G: 2 single dose placebos before surgery + 2 amoxicillin/clavulanate 1000/62.5 mg for 5 days	2 Placebo 1000/62.5 mg single dose before surgery + two placebo 1000/62.5 mg twice daily for 5 days.	N/R	N/R	Postoperative: Metamizol cap one every 8 h, for a minimum of 48 h	Extraction of 3Ms surgery	days 1, 3, and 7 or/and 15 (decided on day 7)	Jan 2022–Dec 2002
Sekhar, Narayanan, and Baig, 2001 [23] India	Double-blind RCT	*n* = 99 19–36 years. F: 44 M: 55	Are prophylactic Ab necessary following the removal of 3Ms, and if so, when and what dose should be given, and for how long?	Metronidazole 400 mg every 8 hourly for 5 days or metronidazole 400 mg orally 8 hourly for five days postoperative	Placebo	1 h before surgery	N/R	Postoperative: Ibuprofen 400 mg tablets as required	Extraction of 3Ms surgery	5 days	N/R
Ataoğlu, Öz, Çandirli, and Kiziloğlu, 2008 [32] Turkey	N/R	*n* = 150 Mean of 20 years. F: 124 M: 26	Evaluate the efficacy of Ab prophylaxis during removal of impacted 3Ms	G1: Amoxicillin/clavulanic acid 1 g twice daily after surgery for 5 days G2: Amoxicillin/clavulanic acid 1 g for 5 days before operation	no prophylaxis given	5 days before surgery	Preoperative rinsed with chlorhexidine mouthwash for 1 min Postoperative: chlorhexidine gluconate mouth rinse twice a day for 5 days	Postoperative: Naproxen sodium 275 mg twice a day for 5 days	Extraction of 3Ms surgery	7 days	Mar 2001–Feb 2003.
Arteagoitia et al., 2015 [29] Europe	Double-blind RCT	*n* = 118 ≥18 years. F: 60 M: 58	Assessing the efficacy of antibiotic for extraction of bone impacted lower 3Ms	Two Amoxicillin/clavulanic acid 1000/62.5 mg tablet 2 h before surgery and every 12 h for 4 days.	Two placebo 1000/62.5 mg tablets every 12 h for 4 days	2 h before surgery	Postoperative chlorhexidine mouthwash	Postoperative: Ibuprofen 600 mg, one sachet 8 hourly as required	Extraction of bone impacted lower 3Ms for any indication	Until week 8 after surgery	Feb 2010–June 2013
Siddiqi, Morkel, and Zafar, 2010 [24] South Africa	Double-blind RCT split-mouth technique	*n* = 95 Mean of 26 years. F: 62 M: 33	Evaluate the potential value of prophylactic antibiotic in 3Ms surgery	GI: 1st visit: Amoxicillin 1 g, 1 h before surgery, 2nd visit: placebo or vice versa.	GII: 1st visit: Amoxicillin 1 g, 1 h before surgery + 500 mg amoxicillin every 8 hourly for 2 days, 2nd visit: placebo or vice versa.	1 h before surgery	Preoperative: chlorhexidine mouthwash Postoperative: chlorhexidine mouthwash 8 hourly for 3 days	Preoperative: Ibuprofen 400 mg Postoperative Ibuprofen 400 mg 6 hourly for 2 days) and paracetamol 500 mg + codeine phosphate 8 mg 6 hourly for 2 days	Extraction of unilateral/contralateral 3Ms surgery	on days 3, 7, and 14 after surgery.	N/R
Bezerra et al., 2011 [25] Brazil	Double-blind RCT	*n* = 34 18–35 years. F: 23 M: 11	Evaluated the effect of prophylactic dosing of antibiotic on the control of postoperative inflammatory/infectious events associated with 3Ms extraction	Two 500 mg amoxicillin Cap	Placebo consisted of 2 capsules (500 mg of starch) each, orally 1 h before the procedure	1 h before surgery	N/R	Postoperative: Nimesulid 100 mg every 12 h for 4 days + dipyrone 500 mg every 6 h for 2 days.	Extraction of all 4 3Ms	after 3, 7, and 14 days	Jan 2008–Nov 2008
Kaczmarzyk et al., 2007 [26] Europe	Double-blind RCT	*n* = 86 18–60 years. F:63 M: 23	Test the hypotheses that (1) administration of a single dose of 600 mg clindamycin 60 min prior to the surgical extraction of a retained lower 3Ms effectively prevents postoperative inflammation compared to placebo, (2) single dose of 600 mg clindamycin prior to surgical extraction of a retained 3Ms with a subsequent 5-day course of 300 mg clindamycin 8 hourly effectively prevents postoperative inflammation compared to placebo	Single-dose group: 600 mg clindamycin hydrochloride 60 min preoperatively, followed by 300 mg placebo every 8 hourly for 5 days.	5-day group: 600 mg clindamycin orally 60 min preoperatively, followed by of 300 mg clindamycin 8 hourly for 5 days.	1 h before surgery	N/R	Postoperative: Ketoprofen in 50 mg cap (Max 200 mg daily).	Extraction of a retained lower 3Ms, (due to orthodontic reasons that required bone removal)	on days 1, 2, and 7 post-operative	Jan 2005–Apr 2006
Xue et al., 2015 [27] China	Double-blind RCT	*n* = 192 18–60 years. F: 118 M: 74	1. To identify whether antibiotics have effect on postoperative inflammatory complications after extraction of impacted mandibular 3Ms in Chinese patients 2. Which antibiotics could be used to prevent and reduce the incidence of postoperative inflammatory complications? 3. When should we give the antibiotic so that it reduces the incidence of postoperative inflammatory complications?	Amoxicillin 0.5 g 1 h preoperatively, or if allergic clindamycin hydrochloride 0.3 g, followed by Amoxicillin 0.5 g, 3 times a day postoperative for 3 days	Placebo 0.5 g (1 h preoperative followed by placebo 0.5 g 3 times a day for 3 days post-operative	1 h before surgery	N/R	Pre and postoperative: Loxoprofen sodium as required	Extractions of bilaterally symmetrical impacted teeth during two visits.	2 and 10 days post-operative	Jan 2013–Dec 2013
Arteagoitia et al., 2005 [34] Europe	Double-blind RCT	*n* = 494 Age: 18–60 years. F: 295 M: 199	1. To assess the efficacy of amoxicillin/clavulanic acid 500/125 mg in preventing infectious and inflammatory complications (IC) in M3 subjects 2. To analyse whether variables such as age, sex, smoking, molar depth, and angulation, ostectomy, odontosection, and intervention time, could be considered risk factors associated with postoperative infectious and inflammatory complications	Amoxicillin/clavulanic acid 500/125 mg oral 3 times a day for 4 days after the intervention	Placebo	N/R	Preoperative: chlorhexidine mouthwash for 1 min Postoperative: chlorhexidine mouthwash 3 times a day for 7 days.	Postoperative: Diflunisal cap 500 mg every 12 h for 2 days as required/For moderate to severe pain: Metamizol cap 575 mg every 8 h	A single M3 was extracted from each patient.	8 weeks	Mar 2001–Feb 2003
Bergdahl and Hedstrom, 2004 [33] Europe	Double-blind RCT	*n* = 119 Age: 17–30 years. F: 67 M:52	The aim of this study was to find out whether metronidazole 1600 mg given as a single dose before extraction of partially impacted mandibular molars reduced the incidence of dry socket	Single dose of metronidazole 1600 mg (four 400-mg tablets),	Placebo	45 min before surgery	N/R	Postoperative: Paracetamol 500 mg with codeine 30 mg	Partially impacted teeth, which had partly broken through the mucosa, and a surgical flap was required to remove the tooth	4 days later	N/R
Luaces-Rey et al., 2010 [28] Europe	Double-blind RCT	*n* = 145 18–60 years. F: 97 M:48	Compare two amoxicillin administration patterns (short prophylactic therapy vs. long postoperative regimen	Short prophylactic therapy: 2 doses in 1 day = 2 g of amoxicillin 1 h before surgery in a single dose	Long postoperative regimen (4 days): 2 tablets of placebo in a single dose 1 h before surgery, followed by a second dose of 1 g of amoxicillin 6 h after and one tablet of 1 g amoxicillin every 8 h for 4 days	1 h before surgery	Postoperative: chlorhexidine mouthwash every 8 h for 7 days	Postoperative: Deflazacort 30 mg one tab every 12 h for 5 days + Dexketoprofen in 25 mg capsule every 8 h as required	Surgical extraction of retained or partially erupted lower third molar due to malposition or any previous infectious or pain episode	7 days post-operatively	N/R
López-Cedrún et al., 2011 [30] Europe	Double-blind RCT	*n* = 123 18–46 years. F: 90 M: 33	Evaluate and compare the effectiveness of 2 different regimens of amoxicillin vs. placebo on the postoperative complications of third molar surgery.	Group A: 4 tablets of amoxicillin 500 mg (total 2 g) 2 h before surgery and 15 tablets of placebo, 3 times daily for 5 days Group B: 4 placebos, 2 h preoperatively and 15 tablets of placebo taken 3 times daily for 5 days.	Group C: 4 placebos 2 h preoperatively and 15 tablets of amoxicillin 500 mg to be taken immediately after surgery 3 times daily for 5 days.	2 h before surgery	Preoperative: a session of professional oral hygiene + chlorhexidine mouth rinse before surgery	Postoperative: Ibuprofen 600 mg immediately after the procedure and every 12 h for 5 days. If not adequate: Metamizol 2 g as required	Patients who had at least one mandibular impacted or partially erupted 3M	7 days	N/R
Rabi et al., 2018 [21] India	Randomized controlled trial	*n* = 60 20–35 years. Gender N/R	Evaluate the effectiveness of antimicrobial therapy in three groups following extraction of an impacted mandibular third molar	Group 1: 625 mg of combined amoxicillin and clavulanic acid tablet for 5 days Group 2: 625 mg of combined amoxicillin and clavulanic acid tablet + 400 mg metronidazole tablet for 5 days	Group 3: no treatment.	N/R	N/R	No description given to the type of analgesics used.	extraction of similar impacted mandibular 3Ms	N/R	N/R
Diz Dios et al., 2006 [20] Europe	Single-double-blinded	*n* = 221 Over 18–57 years. F: 126 M: 95	Evaluate the efficacies of oral prophylactic treatment with amoxicillin (AMX), clindamycin (CLI), and moxifloxacin (MXF) in the prevention of bacteraemia following dental extractions (BDE)	AMX group: prophylactic 2 g of AMX orally 1 to 2 h before anaesthesia induction. CLI group: prophylactic 600 mg of CLI orally 1 to 2 h before anaesthesia induction. MXF group: prophylactic 400 mg of MXF orally 1 to 2 h before anaesthesia induction.	Control group did not receive any type of prophylaxis	1 to 2 h before anaesthesia induction	N/R	N/R	N/R	N/R	Jan 2003–Dec 2004

Total Sample: Total number of participants included in interventions and placebo, i.e., this number is NOT the total number of participants that were recruited at the beginning of the trial, but total number completed the study and reported on, as after recruitment, some participants withdrew from the study for many reasons. N/R: not reported. G: group, F: female, M: male.

**Table 2 medicina-59-00422-t002:** The intervention effect of the selected studies.

Citation	Study Conclusion (Direct Quotation)	Reviewer Comments
Milani et al. (2015) [22]	Pre-operative use of amoxicillin was not effective in reduction of infection, less pain, enabling mouth opening, lymphadenopathy, oedema, and body temperature following removal of fully impacted lower 3Ms in healthy ASA patients.	Only clinically and radiographically evaluated, lower and upper 3Ms impaction, surgeries which involving mandatory osteotomy to access the tooth were included in this study. The study used simple randomisation in which the allocation concealment technique was clearly described. The outcome assessors (surgeon, researcher, and patients) were unaware of group status. The results were standardised by age, gender, participants health condition as well as size, type, and classification of impaction. Amoxicillin was chosen due to its broad antibiotic’s spectrum of activity as well as being first line antibiotic in dentistry. The evaluation performed on the 4th day to allow the finding an infection if occurred. Study limitation: The study was a small study sample of 80 healthy young adults (≥18 years of age up to 30 years), therefore, results may not be generalisable to other population groups.
Mariscal-Cazalla et al. (2021) [31]	Administration of 750 mg amoxicillin before the removal of impacted lower 3Ms in healthy patients with no previous history of infection is unnecessary in reduction of possible complications (pain, inflammation, infection) and postoperative 5-day course of antibiotics is sufficient and postoperative 5-day course of antibiotics is sufficient.	Simple randomisation was used to allocate participants to each intervention. Tooth-related variables were the cause of extraction (preventive extraction, orthodontic cause, or damage of adjacent tooth, osteotomy degree, and coronal section, and the groups did not significantly differ in those variables. Study limitation: This research applied to a specific ethnic community (Caucasian) and therefore may not be generalisable to other populations. Another limitation is that antibiotic was administered 2-h before surgery which might be associated with reduction in the serum peak levels of the antibiotic during surgery, which may not allow for the peak blood levels to be achieved to prevent infection. A further limitation is that they did not include a group that received a preoperative dose alone, which would have revealed whether the same benefits could be achieved with a shorter course of treatment. The inclusion criteria were not fully described and was only standardised to the classification of mandibular third molar impaction. The indication of the surgery was varied between the patients including prophylactic, orthodontic and damage which might influence the result of this study.
Lacasa et al. (2007) [35]	Prophylactic administration of sustained release amoxicillin/clavulanate seems to be well tolerated and been effective in reduction of infection rate, less pain, less swelling, enabling mouth opening in patients undergoing ostectomy.	Groups were homogeneous in most variables recorded in this study and the planned sample size was obtained in all this study. Study limitation: The result of this study should be interpreted with caution due to missing information. This study did not give a definition of antibiotic adverse events, blinding of personnel and blinding of outcome assessment, timing of antibiotic, and the person performing the surgery. No description given on the surgical protocol in this trial. The descriptions of study population, type, position, and classification of third mandibular molar were not provided. Microbiological isolation, and identification of Gram-positive cocci was not carried out. In addition, this study was supported by a grant from GlaxoSmithKline S.A., Tres Cantos, Madrid, Spain.
Sekhar, Narayanan, and Baig (2001) [23]	Administration of pre and postoperative metronidazole 400 mg versus placebo does not seem to add any benefit in reduction of any of the variable measured and no significant differences observed in the outcome between the individual variables of three groups (*p* = 0.09), expect the degree of swelling which was significantly less in the five-day group (*p* = 0.03).	The study had a well-defined research question and standard technique regarding classification of impaction. The reason for choosing metronidazole was appropriately described. Study limitation: There is a concern regard the results of the study where the conclusion was not made based on the outcome being measured, in this case, mortality was not among outcomes measured. The description of study population was not clear. The study includes patients between 19 and 36 years of age making the finding not applicable to all ages. Duration of the study or enrolment period was not specified. The surgical technique was not described in this study.
Ataoğlu, Öz, Çandirli, and Kiziloğlu (2008) [32]	Routine administration of pre and postoperative antibiotic for removal of 3Ms in healthy patients is not supported.	This study compared three different groups: postoperatively for the first group and preoperatively for the second group versus no antibiotic. Presence or absence of infection, swelling, alveolar osteitis, interincisal mouth opening, and pain were evaluated by the same oral surgeon. Study limitations: The number of females participants included in this study is three times greater than number of males which cannot be applied to global population. Study design, setting, or duration, exclusion criteria, and randomisation method were not described, and infection was not defined explicitly. There was no clear explanation of ethics committee approval and how consent was obtained from patients.
Arteagoitia et al. (2015) [29]	Postoperative amoxicillin/clavulanic acid therapy has been more effective for pain relief, reduction in oedema and enabling mouth opening than placebo for extraction of bone impacted lower third molar.	The patients were instructed to self-report their pain, maximum mouth opening, and body temperature and all the post-operative variables were assessed by a single blinded observer. CRP levels were not found to be useful for the diagnosis of infection. The surgical time was measured using a stopwatch from the first incision until the last suture. Study limitation: The mean age between control group and experimental group was statistically significant (*p* = 0.001). The results can only be generalised to Caucasians. Insufficient information was given about exclusion criteria.
Siddiqi, Morkel, and Zafar (2010) [24]	No statistically significant difference in any of variable measured and conclude that administration of preoperative amoxicillin in non-immune-compromised patients versus placebo seems to add no benefit in reduction of infection rate, pain, swelling, trismus, and temperature between the two groups following surgical remove of third molar and routine administration of amoxicillin not recommended.	Each patient acted as their own control using a split-mouth technique. Standard classification of impaction was applied. Study limitation: the number of females included was two times greater than number of males which limit the application of the results to all males. The infection was not stated or defined explicitly. There were eight visits for each patient which potentially affected the compliance for follow-up visits and five patients were unable to complete the follow-up visits for domestic or socioeconomic reasons. No information provided regarding duration of study or time spent in surgery.
Bezerra et al. (2011) [25]	Prophylactic administration of amoxicillin 500 mg does not seem to impose additional benefits on reduction on inflammatory/infectious events to a young, healthy adult population, except for pain and mouth opening in which clear significant difference found between the experimental and control groups.	The reason for choosing the split mouth methodology was explicitly described in this study. Standard technique regarding position and degree of impaction between upper and lower 3Ms of the right and left sides of the mouth was applied according to classification and angulation position was described. In addition, inclusion criteria standardised for age and gender of participants. The mean duration of surgery was homogeneous for the experimental and control groups. Type and extent of the surgical procedures adapted for each case. The reassessment of the patients was performed by the one person. Study limitation: The study was a small cohort of patients, of which most were females. The study included patients with alveolitis and those with history of pain and pericoronitis. Confounders were not controlled or not stated. Participants age between 18 and 35 years, therefore, the results of this study may be more applicable to female younger adults and may not be generalisable to a global population. No information provided regarding the surgical protocol.
Kaczmarzyk et al. (2007) [26]	A single preoperative dose of clindamycin hydrochloride 600 mg with or without subsequent 5-day therapy 600 mg dose made no benefit in reduction of trismus, facial swelling, lymphadenopathy, and postoperative alveolar osteitis in patients following lower third molar surgical extraction with bone removal, except for body temperature in which statistically significant difference found between groups on the seventh postoperative day (*p* = 0.03).	The clindamycin was chosen due to its strong antimicrobial action towards strains isolated from odontogenic infection as well as its ability to reach a high concentration in bone tissue. All surgical procedures were carried out in an identical manner by either one of two oral surgeons, using identical sets of instruments. The duration of sugary, the period between incision of the mucosa and making the last suture been recorded. With regards to the timing of surgery, no statistically significant differences (*p* = 0.48) between groups been revealed. Study limitation: details of the enrolment period were missing. Three patients (3.0%) had clindamycin ceased and replaced by metronidazole due to adverse events; this might affect the results due to intervention inconsistency. There was a lack of clarity in the reporting of adverse events outcome in this study.
Xue et al. (2015) [27]	Pre and postoperative 0.5 g amoxicillin versus placebo made no significant effect to prevention of postoperative IC after extraction of bilaterally symmetrical impacted mandibular 3Ms and do not contribute to better wound healing or increased mouth opening, except for pain after extraction score for pain on day 10 (*p* = 0.005).	The definition of impacted teeth was well defined and classified. This study was well designed as split-mouth. Moreover, the mean extraction time was no longer than 30-min in each group. Study limitation: This study included alveolar osteitis as a sign of postoperative inflammation and infection. The infection was not stated or defined explicitly in sufficient detail in this study. There was a lack of clarity in the reporting of adverse events and criteria of postoperative complications and adverse reactions been mixed and reported as postoperative inflammation, including specific infection and adverse reactions. The results can be generalised to a specific Chinese population. There was lack of clarity in reporting adverse events due to antibiotics or infection and other complications. The dose regimen was not standardised and patients allergic to the used antibiotics received a modified regimen of another antibiotic which may hinder the results accuracy.
Arteagoitia et al. (2005) [34]	Supported the use of postoperative amoxicillin/clavulanic acid in 3Ms partially covered by bone and those in a horizontal position	Groups (placebo and antibiotic) were well balanced with respect to age, sex, smoking, molar depth, molar angulation, third molar position, intervention time, and need for sectioning. However, ostectomy was greater in placebo group. Study limitation: Dry socket included as a diagnosis of postoperative infection and inflammatory complication. The timing of postoperative administration antibiotic was not reported.
Bergdahl and Hedstrom (2004) [33]	A single dose of preoperative metronidazole was pointless in prevention of postoperative alveolar osteitis in those require removal of partially impacted teeth and does not seem to be beneficial in this case.	The inclusion criteria were only standardised for duration of operation, amount of saline irrigation, and amount of bone removed, both depth and volume. Study limitation: The participants were between the age of 17 and 30 years; therefore, the results of this study may be more applicable to younger adults than the global population. There was insufficient reporting on the inclusion and exclusion criteria. Female participants were taking contraceptive medication or had their menstrual periods during the procedures, and some (males and females) had history of pericoronitis or smoking tobacco which may have affected the patients’ outcomes. This study did not report the duration of study.
Luaces-Rey et al. (2010) [28]	Pre-and post-operative use of amoxicillin did not contribute better haematoma appearance, wound healing, less infection and intraoral inflammation, increased trismus and does not seem to reduce intake of rescue analgesics for pain management.	Patients were randomly enrolled to one of the two treatments according to the type of antibiotic treatment (short prophylactic antibiotic regimen or long postoperatively regimen). Patient, surgeon, and observer were unaware of intervention groups during the whole study. Both groups were homogeneous in every evaluated parameter including preoperative radiographic and clinical examination such as the tooth to be extracted (left, right) and the degree of impaction, position, and maximum preoperative oral opening between incisions. Each patient was provided with a form to self-report different numeric score for each outcome; pain score, type of diet (liquid, soft, or normal), corporal temperature, number of analgesic tablets taken every day, trismus (a 5 mm reduction was considered clinically significant). Each procedure was timed from first incision to completion of last suture. Study limitation: The number of females was two times greater than males making the results less applicable to the global population. The details of the duration of the study were not provided. The authors included a variety of definitions for infection using different clinical parameters which may also affected their conclusion.
López-Cedrún et al. (2011) [30]	Post-operative amoxicillin therapy versus placebo leads to lower infection, less fever and dysphagia, except intra and extraoral swelling which antibiotic does not seem to be effective in lowering intra and extraoral swelling following mandibular impacted or partially erupted third molar.	The number of ostectomies performed was similar in the three groups. In addition, the operative time was similar in all three groups. The standard surgical procedure been carried out and only one lower 3M was removed at a time. The mean operative time was longer in the infected patients (277 s) than non-infected patients (239 s). The two hours before the surgical procedure was to ensure the therapeutic antibiotic tissue level was reached before surgery. Study limitation: The participants were between the age of 18 and 46 years old; and the number of females included was three times more than males included in the study, therefore, the results may be more applicable to younger female adults, not the global population. The length of a clinical study was not specified.
Rabi et al. (2018) [21]	Administration of postoperative antibiotic was not effective in reduction of postoperative clinical infection evaluated between the individuals of all the three study groups following the extraction of impacted mandibular third molar.	There was no statistically significant difference between the mean ages of all study groups. Second group show slightly better satisfaction than other groups. Study limitation: The study was of a small sample. Timing of the antibiotic administered was not described. The infection was not stated or defined in sufficient detail. The authors did not explain whether the study was blinded or not. The study may be more applicable to younger adults; therefore, results may not be generalisable to the global population. Duration of surgery was missing in this study. This study is lacking relevant outcomes measured in this review.
Diz Dios et al. (2006) [20]	Amoxicillin and moxifloxacin prophylaxis showed significantly high efficacies (*p* < 0.001 for both) in reducing the prevalence and duration of bacteraemia whereas clindamycin prophylaxis was noneffective.	The study comprised of patients who, for behavioural reasons (autism, learning disabilities, phobias, etc.), underwent dental extractions under general anaesthesia. No significant differences were found between the different study age, sex, oral health grade, or number of teeth extracted groups. The incidence of bacteraemia was evaluated using peripheral venous blood sample taken from each patient before the dental manipulation and 30-s, 15-min, and 1-h after the final completion of the dental extractions. Study limitation: No description provided on who performed the surgery and the surgical technique was not described. This study did not describe the reasoning for dental extraction, or the duration of surgery and the participants’ inclusion and exclusion criteria were missing. None of this review’s measured outcomes was reported.

**Table 3 medicina-59-00422-t003:** Effective Public Health Practice Project (EPHPP) quality assessment tool rating for individual studies.

Citation	Selection Bias	Study Design	Confounders	Blinding	Data Collection Methods	Withdrawals and Dropouts	* Global Rating (Mean Score)
Milani et al., 2015 [22]	3	1	0	1	2	1	1
Mariscal-Cazalla et al., 2021 [31]	1	1	1	1	2	1	1
Lacasa et al., 2007 [35]	1	1	1	2	2	1	1
Sekhar, Narayanan, and Baig, 2001 [23]	1	1	0	1	2	2	1
Ataoğlu, et al., 2008 [32]	3	3	0	3	2	3	2
Arteagoitia et al., 2015 [29]	1	1	1	1	2	1	1
Siddiqi, Morkel, and Zafar, 2010 [24]	1	1	0	1	2	1	1
Bezerra et al., 2011 [25]	2	1	0	1	2	1	1
Kaczmarzyk et al., 2007 [26]	1	1	1	1	2	1	1
Xue et al., 2015 [27]	1	1	1	1	2	1	1
Arteagoitia et al., 2005 [34]	1	1	1	1	2	1	1
Bergdahl and Hedstrom, 2004 [33]	1	1	1	1	2	1	1
Luaces-Rey et al., 2010 [28]	1	1	1	1	2	1	1
López-Cedrún et al., 2011 [30]	2	1	1	1	2	1	1
Rabi et al., 2018 [21]	1	2	1	2	2	1	2
Diz Dios et al., 2006 [20]	1	1	1	2	2	1	1

* Strong = 1, Moderate = 2, Weak = 3, NA: Confounders not controlled for, or not stated = 0, Studies with strong (1) or moderate (2) global mean score were deemed suitable for the systematic review. Selection bias: Strong = participants very likely to be representative of the target population of greater than 80% participation. Moderate = participants very likely to be representative of the target population of 60–79% participation. Weak = participants very likely to be representative of the target population of less than 60% participation. Study design: Strong = randomised controlled trial (RCTs) or controlled clinical trial (CCTs). Moderate = cohort analytic study, a case control study, a cohort design, or an interrupted time series. Weak = any other method or did not state the method used. Confounders: Strong = controlled for at least 80% of relevant confounders. Moderate = controlled for 60–79% of relevant confounders. Blinding: Strong = outcome assessor not aware of intervention status of participants AND participants not aware of research question. Moderate = outcome assessor is not aware of the intervention status of participants OR the study participants are not aware of the research question. Weak = outcome assessor AND participants both aware of the above or blinding is not described. Data collection methods: Strong = data collection tools shown to be valid AND reliable. Moderate = Tools are valid, but reliability not described. Weak = data collection tools not shown to be valid. Withdrawals and dropouts: Strong = Follow-up rate more than 80% or greater. Moderate = Follow-up rate of 60–79% of participants. Weak = Follow-up rate less than 60% participants or withdrawal and dropouts were not described. Weak = controlled for less than 60% relevant confounders.

**Table 4 medicina-59-00422-t004:** Factors association with increased the risk of infections.

Citation	Increased Risk of Infection	Yes/No	Total
Arteagoitia et al., 2005 [34]	Age	Yes	2, 9.5%
López-Cedrún et al., 2011 [30]	Age	Yes	-
Milani et al., 2015 [22]	Clinical symptoms	Yes	1, 5%
Luaces-Rey et al., 2010 [28]	Difficult surgery	Yes	1, 5%
López-Cedrún et al., 2011 [30]	Gender	Yes	1, 5%
Arteagoitia et al., 2015 [29]	Inflammation	Yes	1, 5%
López-Cedrún et al., 2011 [30]	Length of procedures	No	1, 5%
Lacasa et al., 2007 [35]	Length of procedures	Yes	2, 9.5%
Luaces-Rey et al., 2010 [28]	Length of procedures	Yes	-
Sekhar, Narayanan, and Baig, 2001 [23]	Placebo	No	5, 24%
Siddiqi, Morkel, and Zafar, 2010 [24]	Placebo	No	-
Xue et al., 2015 [27]	Placebo	No	-
Bezerra et al., 2011 [25]	Placebo	No	-
Arteagoitia et al., 2015 [29]	Placebo	No	-
Lacasa et al., 2007 [35]	Placebo	Yes	2, 9.5%
Ataoğlu, et al., 2008 [32]	Placebo	Yes	-
Mariscal-Cazalla et al., 2021 [31]	Requiring rescue ABs post-surgery	Yes	1, 5%

**Table 5 medicina-59-00422-t005:** Factors association with postoperative facial oedema or swelling.

Citation	Antibiotics vs. Placebo	Total
Mariscal-Cazalla et al., 2021 [31]	Favour ABs	2 (12%)
Sekhar, Narayanan, and Baig, 2001 [23]	Favour Abs	
López-Cedrún et al., 2011 [30]	No difference	7 (44%)
Siddiqi, Morkel, and Zafar, 2010 [24]	No difference	
Xue et al., 2015 [27]	No difference	
Lacasa et al., 2007 [35]	No difference	
Ataoğlu, et al., 2008 [32]	No difference	
Kaczmarzyk et al., 2007 [26]	No difference	
Milani et al., 2015 [22]	No difference	

**Table 6 medicina-59-00422-t006:** Findings comparison with previous systematic review.

Ramos et al., 2016 [42]	With regard to other published meta-analyses, Ramos et al. (2016) assessed the efficacy of any orally or parenterally administered antibiotics at any dose or regimen, and with regard to comparisons in preventing dry socket and/or infection after third molar extraction. The author included 21 double-blinded RCTs analysed for dry socket and/or infection and 12 studies analysed for adverse events. The author found that antibiotics administered to prevent dry socket and/or infection are beneficial in reducing the risk of infection after third molar extraction by 57% (RR = 0.43; 95% CI 0.33–0.56; *p* < 0.0001) with NNH of 16 patients. This is comparable with the findings of the current meta-analysis, which found a higher reduction of infection by 69% with NNT of 17. In addition, this review did not combine dry socket with infection in the metanalysis, they were analysed separately. However, with regard to adverse events, the findings are different from the current findings where the risk of adverse reactions from AB administration was found not statistically significant with a smaller NNH of 16 patients. In this metanalysis, the NNT was 55 patients but this metanalysis was limited to RCT studies published since 2000 in which only oral systemic antibiotic use was analysed.
Singh Gill et al., 2018 [43]	The systematic review and metanalysis performed by Singh Gill et al. (2018) identified four RCTs published since 2000 until 2013 in their final review to evaluate the effectiveness of antibiotic prophylaxis in preventing infections in extraction procedures in patients requiring single/multiple dental extractions for various indications (impacted wisdom teeth, abscess etc.) undergoing dental extraction and the results showed that there was no clear evidence pointing to the need for prophylactic antibiotics for reduction of infection after third molar extraction. The current findings and recommendations were not congruent with theirs, where a significant reduction of infection with NNT of 17 patients was identified, although the current analysis included a recently published study with a large number of patients. The systematic review and metanalysis conducted by Singh Gill et al. (2018) included three randomised controlled trials (RCTs) which compared the administration of any antibiotic vs. placebo to patients undergoing dental implant placement. The author concluded that prophylactic antibiotics were not beneficial in those undergoing implant surgeries since *p* value (*p* = 0.09) with NNT of 33 under normal conditions. This finding differs from the current study, The variation in conclusions is related the inclusion of three additional studies in this analysis. A greater (65%) reduction in risk of implant failure with antibiotic with a smaller NNT was observed in this metanalysis. In addition, the study concluded that there is lack of evidence that antibiotic use reduces the risk of post-operative complications following dental implant placement. These findings and recommendations were similar to theirs, although we included two additional RCTs in our analysis.

## Data Availability

Not applicable.

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
