# Peer review of "The Role of Antibiotic Use in Third Molar Tooth Extractions: A Systematic Review and Meta-Analysis"

_medicina, 2023, doi:10.3390/medicina59030422_

Round 1
Reviewer 1 Report
The article is a systematic review and meta-analysis investigating the role of antibiotics in reducing postoperative infections after third molar tooth extractions. The reported systematic review appears to be well designed and adheres to accepted guidelines and best practices in systematic review methodology, such as PRISMA-P methdology and the Cochrane Handbook. The use of the Cochran-Mantel-Haenszel method for combining dichotomous data and reporting the relative risk with 95% confidence intervals is appropriate. The use of the Review Manager software and performing a sensitivity analysis when heterogeneity was detected are also commendable. The search strategy appears comprehensive and makes use of appropriate medical subject headings and key words to search the PubMed database. The eligibility criteria for studies appear to be clearly defined and appropriate.
However, there are some critical comments:
1) The study only searched for evidence in three specific databases (PubMed, Science Direct, and the Cochrane Database), which may not have captured all relevant studies in the field. The authors should have considered searching for evidence in additional databases and other sources, such as conference proceedings, to ensure that the systematic review was as comprehensive as possible.
2) The study only included English language full-text publications, which could lead to publication bias.
3) Three clinical trials with high risk of bias were included (Ataoglu et al, Bezerra et al, and Lacasa et al), and only 4 clinical trials were at low risk of bias in all domains using the Cochrane risk-of-bias version.
4) Clinical trials with different antibiotics (amoxicillin, amoxicillin with clavulanic acid,clindamycin, methotrexate) and with different dosing regimens were included. Although the most evaluated antibiotic was amoxicillin at a dose of 1 g, we do not know if there are differences in effects between the different antibiotics and with different dosing regimens.
5) The study only considers the efficacy and safety of different antibiotic regimens versus a placebo, and does not take into account other factors that may influence the outcome of postoperative infections (age of patients, comorbidity, surgical technique, length of procedure, different healthcare system).
Overall, the systematic review appears to be well designed, but the authors should comment on the limitations mentioned above related to the validity of the results.
Author Response
|
Reviewer 1 |
|
|
The study only searched for evidence in three specific databases (PubMed, Science Direct, and the Cochrane Database), which may not have captured all relevant studies in the field. The authors should have considered searching for evidence in additional databases and other sources, such as conference proceedings, to ensure that the systematic review was as comprehensive as possible. |
Added to limitations |
|
The study only included English language full-text publications, which could lead to publication bias. |
Authors and publishers were approached to provide English version , complete text or additional data, however, in the cases where response was not received or received informing with unavailability, the study was not possible to be included. The research was low cost and there was not ability to pay for translating text. |
|
Three clinical trials with high risk of bias were included (Ataoglu et al, Bezerra et al, and Lacasa et al), and only 4 clinical trials were at low risk of bias in all domains using the Cochrane risk-of-bias version. |
Not sure where this was reported, only one study had one element of high risk of bias. |
|
Clinical trials with different antibiotics (amoxicillin, amoxicillin with clavulanic acid, clindamycin, methotrexate) and with different dosing regimens were included. Although the most evaluated antibiotic was amoxicillin at a dose of 1 g, we do not know if there are differences in effects between the different antibiotics and with different dosing regimens. |
Comparing antibiotics or selecting studies used the same antibiotic was not the intention of the review, the review intended to compare the use of antibiotics vs. no antibiotics use. |
|
The study only considers the efficacy and safety of different antibiotic regimens versus a placebo and does not take into account other factors that may influence the outcome of postoperative infections (age of patients, comorbidity, surgical technique, length of procedure, different healthcare system). |
Yes, was included in the narrative review, as it was not possible to conduct statistical analysis due to the type of data reported in the included studies. |
Reviewer 2 Report
I would like to congratulate the authors for conducting the present review regarding the use of antibiotics on the third molar teeth extraction. I would like to share a few concerns:
The abstract has subheading, however they are not recommended according to the journal guidelines.
I suggest the authors to place the keywords by alphabetic order
I recommend the authors to improve a little better the rationale of the study. Why do we need this paper that may justify to publish it? Is there any gap in the knowledge that justify this study? Does this study intends to close that gap?
I also suggest the authors to improve the aim sentence in the end of the introduction. May the authors be clearer regarding what are their intentions and outcomes to assess?
I don’t understand the subheading “Review Rationale”. This subheading has two sentences. The first is not part of a rationale to conduct a study. While the second is a rephrasing of the last sentence of the introduction.
Regarding the “Methods and Design” this is not a subheading according to the journal guidelines.
The statistical analysis should come in the end of the material and methods. First the authors should describe the objectives, variables, how they assessed them and only in the end how they compared them in the statistical analysis.
I recommend the authors to review the inclusion/exclusion criteria. The authors have placed multiple topics on the exclusion criteria that are solo the opposite of items placed in the inclusion list. The exclusion criteria is not necessary the opposite of the inclusion (those are already in the inclusion), they are instead possible source of bias.
The authors do not mention the screening method in detail.
Did the authors conduct a manual search?
Did the authors contact authors with publications in the field?
Did the authors search grey literature?
Why did not the authors conducted the mandatory (according to Prisma) risk of bias assessment? Was this replaced by what the authors name “quality assessment”? If yes, how many observers did it? May the authors provide the workflow for this item?
Why did the authors only search in PUBMED when is it recommended to conduct at least in two electronic databases?
Did the authors use any filters or limits?
What were the dates from when to when the search was conducted?
From which dates were the articles the authors aimed for? Any date range?
I would recommend the authors to split the “Results” and “Discussion” in 2 different sub-sections, and not combine them?
What was the review level of evidence?
I do not understand the need for a sub-heading named “Reviewer Discussion”. Who is the “reviewer”? Is this the Discussion? If yes… why do we have before a “Results and Discussion”? I would recommend to join the Discussions
Is the “Reviewer Conclusion” the “Conclusion”? If yes, I suggest to remove the word “Reviewer”
Author Response
|
Reviwer 2 |
|
|
The abstract has subheading; however they are not recommended according to the journal guidelines. |
Corrected |
|
I suggest the authors to place the keywords by alphabetic order |
Corrected |
|
I recommend the authors to improve a little better the rationale of the study. Why do we need this paper that may justify publishing it? Is there any gap in the knowledge that justify this study? Does this study intend to close that gap? |
Corrected |
|
I also suggest the authors to improve the aim sentence in the end of the introduction. May the authors be clearer regarding what are their intentions and outcomes to assess? |
Corrected |
|
I don’t understand the subheading “Review Rationale”. This subheading has two sentences. The first is not part of a rationale to conduct a study. While the second is a rephrasing of the last sentence of the introduction. |
Corrected |
|
Regarding the “Methods and Design” this is not a subheading according to the journal guidelines. |
Corrected |
|
The statistical analysis should come in the end of the material and methods. First the authors should describe the objectives, variables, how they assessed them and only in the end how they compared them in the statistical analysis. |
Corrected |
|
I recommend the authors to review the inclusion/exclusion criteria. The authors have placed multiple topics on the exclusion criteria that are solo the opposite of items placed in the inclusion list. The exclusion criteria is not necessary the opposite of the inclusion (those are already in the inclusion), they are instead possible source of bias. |
Corrected |
|
The authors do not mention the screening method in detail. |
Corrected |
|
Did the authors conduct a manual search?
|
All studies found by manual search were also found as duplicated in the actual search, as such they were not specifically separated |
|
Did the authors contact authors with publications in the field? |
Yes, see exclusion criteria |
|
Did the authors search grey literature?
|
Yes, but for the introduction and discussion sections. |
|
Why did not the authors conduct the mandatory (according to Prisma) risk of bias assessment? Was this replaced by what the authors name “quality assessment”? If yes, how many observers did it? May the authors provide the workflow for this item? |
Risk of bias included. Statement about the process is now added. |
|
Why did the authors only search in PUBMED when it is recommended to conduct at least in two electronic databases? |
Added to limitation |
|
Did the authors use any filters or limits?
|
Yes, included under search strategy |
|
What were the dates from when to when the search was conducted? From which dates were the articles the authors aimed for? Any date ranges? |
In the selection criteria: January 2000 -November 2021 |
|
I would recommend the authors to split the “Results” and “Discussion” in 2 different sub-sections, and not combine them? |
Corrected |
|
What was the review level of evidence? |
Corrected |
|
I do not understand the need for a sub-heading named “Reviewer Discussion”. Who is the “reviewer”? Is this the Discussion? If yes… why do we have before a “Results and Discussion”? I would recommend joining the Discussions |
Corrected |
|
Is the “Reviewer Conclusion” the “Conclusion”? If yes, I suggest removing the word “Reviewer” |
Corrected |
Round 2
Reviewer 2 Report
Dear authors, I have no more concerns.